# Effect of the Use of Gnrh Analogs in Low-Grade Cerebral Glioma

**DOI:** 10.3390/children10010115

**Published:** 2023-01-05

**Authors:** Ana de Lucio Delgado, Jose Antonio Villegas Rubio, Isolina Riaño-Galán, Juan Pérez Gordón

**Affiliations:** 1Oncology Pediatric Department, Central University Hospital of Asturias, 33011 Oviedo, Spain; 2Pediatric Endocrinology Department, Central University Hospital of Asturias, 33011 Oviedo, Spain

**Keywords:** low-grade glioma, child, steroid hormones

## Abstract

Low-grade gliomas are the most common brain tumors in children. This tumor type presents a wide range of clinical, histological, and biological behaviors. In recent years, an association between estrogens and progesterone and the development of tumors has been suggested. A case of a 2-year-old girl is described with a low-grade brain tumor treated with chemotherapy and disease stabilization. The treatment with Decapeptyl® was initiated due to precocious puberty, and the tumor showed a decrease in its solid component—more than 50% of the initial size—three years after starting treatment. Several studies have described the influence of estrogen and progesterone on the development of gliomas, decreasing or increasing their expression in those tumors with greater aggressiveness, respectively. Despite the fact that the tumor-hormonal expression relationship in other tumor types has been evaluated, its role in the treatment of brain tumors remains unknown.

## 1. Introduction

Brain tumors are the second most common cause of childhood cancer in children between 0 and 14 years of age (approximately 22%) and the third most common cause of cancer in adolescents between 15 and 19 years of age (close to 10% of the total) [1]. Low-grade gliomas (LGG) represent the most common subgroup (30–50%) and can affect any location in the central nervous system (CNS) [2]. According to the 2021 WHO classification, around 75% of all histologically confirmed gliomas in children are LGG (grades I and II). In this classification, tumors are also categorized, by their latest molecular characteristics (genotypic) in addition to their histological characteristics (phenotypic) [3].

The LGG constitute a heterogeneous group of tumors regarding clinical symptoms, histology, and biological behavior. Up to 5–10% can present metastases in other locations at diagnosis [4]. The pilocytic astrocytoma (with its characteristic Rosenthal fibers) is the most frequent histological subtype in children, accounting for 20% of the total [5]. Despite the fact that this tumor arises predominantly in the cerebellum, optic pathways, and brain stem, it can be found anywhere in the CNS. The most frequent location of LGG [6] is shown below (Figure 1).

The annual incidence of LGG is close to 10–12 cases per million children under fifteen years of age, with a male-to-female ratio of 1.1:1/1.2:1. Although the symptoms may vary, most are indolent and do not transform into malignant tumors (even cases of spontaneous regression have been described) [7]. On the contrary, LGG in adults presents different biology and behavior, impacting prognosis and overall survival.

Estrogens are steroid hormones that affect different systems: reproductive, gastrointestinal, skeletal, immune, and CNS. Most of its effects are mediated by alpha (ER-α) and beta (ER-β) estrogen receptors [8,9]. Several studies have demonstrated that the CNS effects of estrogens are not only limited to the resolution of vasomotor instability; they are also extended to psychological disturbances [10] (depression, behavioral changes, or cognitive dysfunction) and the development of tumors [11]. The receptors alpha and beta are expressed under physiological conditions and are lost or reduced during tumor development, indicating a potential tumor suppressive function [12]. Progesterone is also involved in regulating the reproductive system, including ovulation and sexuality. In addition, it influences neuronal excitability, learning, and the neoplastic proliferation of glial cells. These effects result primarily from the interaction between hormone-progesterone receptor alpha (PR-α) and hormone-progesterone receptor beta (PR-β). Both PRs modify the expression of genes involved in cell proliferation, angiogenesis, and epidermal growth factor (EGF) production [13].

This manuscript describes our experience with a pediatric patient with an LGG and her response to gonadotropin-releasing hormone (GnRH) analog drugs.

## 2. Case Report

In November 2017, a two-year-old boy and a one-month-old girl were consulted in the Pediatric Emergency Department of a tertiary-level hospital due to persistent vomiting, gait disturbances, and noticeable instability after one week of evolution. Given this clinical picture, a head computer tomography was performed, revealing a mass in the supratentorial region with a significant component of hydrocephalus. The study was completed with cranial magnetic resonance imaging (MRI) (a single mass located in the suprasellar region with a solid and cystic component was found). The hormonal studies (FSH, LH, estradiol, progesterone, prolactin, ACTH, cortisol, IGF-I, IGFBP3, TSH, FT4, and FT3), tumor markers (AFP and B-HCG), and visual and auditory potentials (all these last tests did not show any alteration) were conducted. A ventriculoperitoneal (VP) shunt was placed to improve the symptoms of intracranial hypertension, and a biopsy of the lesion was performed during the same surgical act. A pathological report confirmed the etiology of pilomyxoid astrocytoma (LGG, grade I) with a KIAA1549-BRAF mutation. In December 2017, chemotherapy based on weekly Vinblastine (6 mg/m^2^/dose) was initiated for 70 weeks, according to the recommendations of “*Vinblastine in recurring or refractory pediatric low-grade glioma*” [14], with excellent tolerance and no dose reduction required at any time. The patient finished the treatment in May 2019 and did not present any neurological symptoms at any time.

In addition, no changes were observed throughout the imaging controls carried out every 3 months in terms of the volume of the lesion or its contrast uptake (except for a slight initial increase of the mass in the first months, which can be explained by the antiangiogenic activity of vinblastine that can cause an initial growth during the first months of treatment followed by a subsequent reduction in tumor size [15]). The tumor mass was described at the end of treatment with similar measurements compared to the size at the start of treatment. The hormonal analytical controls requested every 3–4 months throughout the chemotherapy process showed values within the normal range. However, two years and one month after the start of Vinblastine (January 2019), an increase in LH was observed, rising from 1.9 U/L(normal range < 0.1–0.5 U/L) to 6.7 U/L four months later. FSH values also increased from 5.6 U/L to 7.5 U/L (normal range 0.2–11.1), estradiol from 8.6 pg/mL to 17.3 pg/mL (normal range for the patient’s age < 5 pg /mL) and progesterone from 0.05 to 0.46 ng/mL (normal range for the patient’s age < 0.05 ng/mL) (Figure 2 and Figure 3). The rest of the hormonal study (thyroid hormones, prolactin, DHEA, androstenedione, total testosterone, free testosterone, ACTH, and cortisol) as well as the blood and biochemistry count presented values within the established parameters. However, the values of IGF-1 and IGFBP3 were slightly above the upper range of normality. The anthropometric data, physical examination, and other complementary tests of the patient in May 2019 were as follows: Weight: 20.8 Kg (+1.11 SD), Height: 11 4.5 cm (+2.06 SD); TA: 99/53 (p59/p40); Growth rate: 19.6 cm/year (+15.13 SD), with 2.2 cm of growth in the last month and a half. The presence of a bilateral breast button was detected, not axillary, at stage I pubarche. Normal female external genitalia. Bone age: close to 6 years of age at birth for a chronological age of 4 years and 4 months. The pelvic ultrasound performed to assess the sexual appendages was concordant with the patient’s age. The studies were completed with the LH-RH test, which showed values compatible with activation of the pituitary-gonadal axis (FSH at 0 min of 4.8 U/L, 30 min of 18 U/L, and 60 min of 17.9 U/L, all within normal range; LH at 0 min 3.8 U/L, 30 min: 50.5 U/L and at 60 min 36.8 U/L). In addition, given the reported data of precocious puberty, treatment with triptorelin, a gonadotropin-releasing hormone analog, was started in July 2019 at a dose of 300 μg/Kg dose every 12 weeks. Despite the practical normalization of the FSH and LH values, the patient continued to have a striking growth rate, requiring dose adjustment to 128 μg/Kg/dose every 21 days.

In the control MRI performed two months after starting treatment with triptorelin, a decrease in the solid component of the tumor was observed, along with a significant decrease in contrast uptake. In successive controls, this decrease resulted in more than 50% less compared to the initial size without contrast uptake (last image, June 2022) (Figure 4). The Pathological Anatomy Service was requested to review the preserved tumor tissue from the diagnosis to verify the presence or absence of estrogen and progesterone receptors using immunohistochemical techniques, resulting in negative results.

Regarding hormonal controls, IGF-1 and IGFBP3 values showed a progressive increase, with an altered response to repeated oral glucose overload tests. Given the suspicion of failure in the growth hormone axis that could explain the non-control of growth in our patient with the use of triptorelin, treatment with Octreotide acetate, 1/2 vial i.m., was started in September 2021 every 28 days, being able to space triptorelin every ten weeks. Latest anthropometric data and physical examination at seven years and five months: Weight: 32.5 kg (+1.15 SD), Height: 138.6 cm (+2.66 SD), Growth rate: 4.7 cm/year (p18, −0.95 SD), BP 97/65 (p30/p63). Thelarche 1, not Axilarche ubic hair on the labia majora, not on the mount of Venus.

## 3. Discussion

The steroid hormones have a prominent neuroprotective role in neurological disorders, such as Parkinson’s or Alzheimer’s disease (improving myelination, decreasing inflammation and edema, and inhibiting apoptosis), due to their antioxidant properties or activation [12]. However, in 1983, the potential relationship between these hormones and the development of brain tumors was first reported [16].

Several studies have described the influence of estrogens on the development of gliomas through the direct interaction with their intracellular receptors ER-α and ER-β [17,18,19] (which are ligand-activated transcription factors that regulate the expression of several genes involved in reproduction, development, metabolism, and cell proliferation) or, since steroid hormones are members of a superfamily of transcription factors, through the activation of potentially oncogenic mediators [20]. In addition, both ER isoforms are expressed in astrocytomas, with a clear predominance of ER-α. These receptors are highly homologous despite being products of different genes; ER-α is located on chromosome 6q25.1, and ER-β is situated on chromosome 14q22–24 [9]. At least five ER-β (ER-β 1–5) isoforms have been identified [21]. While ER-α has been extensively investigated in various neoplasms, the specific role of ER-βin the pathogenesis, progression, and prognosis of these neoplasms remains unknown [22]. It has been suggested that the loss of expression of this receptor could be an essential step in estrogen-dependent tumor progression. The presence of both ERs decreases in those astrocytomas with higher grades of malignancy, thus suggesting their neuroprotective role [23]. These findings, together with the higher incidence of gliomas in men and postmenopausal women, suggest that estrogens could reduce tumor proliferation [24]. However, several studies published by González-Arenas et al. have shown that “in vitro” estradiol induces astrocyte growth through its interaction with ER-α and its regulation of gene expression in the cell cycle, angiogenesis, and metastasis [25,26]. Selective estrogen receptor modulators could explain all these results (SERMs), which can function as estrogen antagonists or agonists, depending on the target tissue and ER subtype, thus mediating a specific response. In this way, individual SERMs can act as pure agonists, pure antagonists, or mixed agonists/antagonists [27].

Progesterone plays a neuroprotective role after an injury to the CNS and peripheral nervous systems, limiting tissue damage or improving the functional prognosis after traumatic brain injuries, strokes, spinal cord injuries, and diabetic neuropathy, among others [28]. This hormone crosses the blood-brain barrier, decreasing the inflammatory process and edema [29]. Although the actual mechanisms responsible for these effects remain unknown, it is believed that the major cause is the synthesis and stimulation of secretion of neuroprotective substances: neuronal growth factor (NGF), brain-derived neurotrophic growth factor (BDNF), or glial cell line-derived neurotrophic factor (GDNF) [30].

Furthermore, progesterone also has a known role in the proliferation of brain tumors [13]. Its receptors have been found in several types of brain tumors, such as chordomas, craniopharyngiomas, and gliomas [31]. However, the best-known association is with meningiomas [32]. This tumor occurs twice as often in women as in men [33]. It is known that meningiomas can grow during pregnancy and menstruation (both situations in which progesterone levels are high) as well as have a positive association with breast tumors. The molecular and immunohistochemical studies confirm that meningioma is a hormone-sensitive tumor, with approximately 70% of meningiomas ex-pressing PR and approximately 30% expressing ER [34]. The presence of PR in meningiomas carries a better prognosis and a lower probability of recurrence. Similar receptors have been found in glial tumors. PR-α is predominant in meningiomas, while PR-β is more frequent in gliomas. In addition, PR expression increases with the histologic malignancy of astrocytomas. “In vitro” studies indicate that progesterone promotes cell proliferation in these tumors, as well as the expression of genes that are important for tumor growth and spread [35]. On the contrary, similarly to estrogen properties, other studies indicate that progesterone also has antiproliferative properties and apoptotic effects in ovarian, breast, endometrial, and colon tumors, as well as in gliomas [36].

In our patient, the association between the initiation of triptolerin and the decrease in the solid component of the tumor seems clear, despite the expression of negative markers in the sample studied. This could be because ER and PR expression is heterogeneous in the tumor tissue and has not been detected due to the scarcity of the sample (diagnostic biopsy, not tumor resection). In addition, although the effect of chemotherapy in reducing the tumor size can be observed sometime after its initiation, in our case the relationship with the observed response more than one year after the treatment initiation without dose reduction is improbable. Moreover, some cases of spontaneous regression in children with low-grade gliomas have been reported, in contrast to adult patients who tend to show more aggressive behavior [37,38]. In our case, there is an important correlation in time between the establishment of the treatment with triptorelin and the decrease in tumor size, which might indicate a strong relationship between these two factors.

## 4. Conclusions

The treatment with triptorelin in this patient led to an evident reduction in the size of the tumor that was not achieved with chemotherapeutic treatment. Moreover, this treatment had no side effects and was well tolerated. To date, this is a promising response, although the outcome after the completion of therapy is unknown. Thus, more extensive studies in the pediatric population are needed to obtain more reliable data on the effectiveness of hormonal treatment for this type of tumor, assess the situation once treatment is discontinued, and also evaluate the development of long-term side effects.

## Figures and Tables

**Figure 1 children-10-00115-f001:**
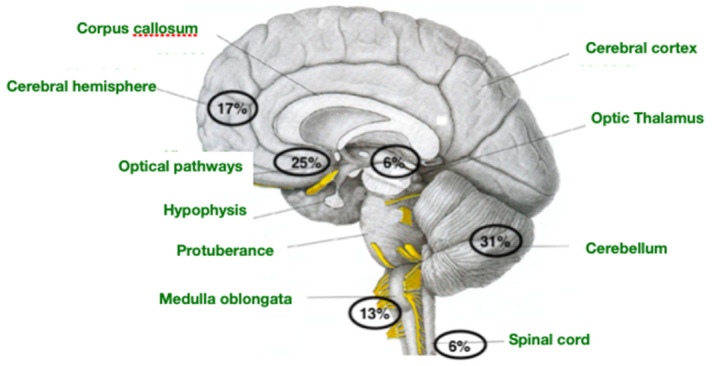
Most frequent location of low-grade glioma.

**Figure 2 children-10-00115-f002:**
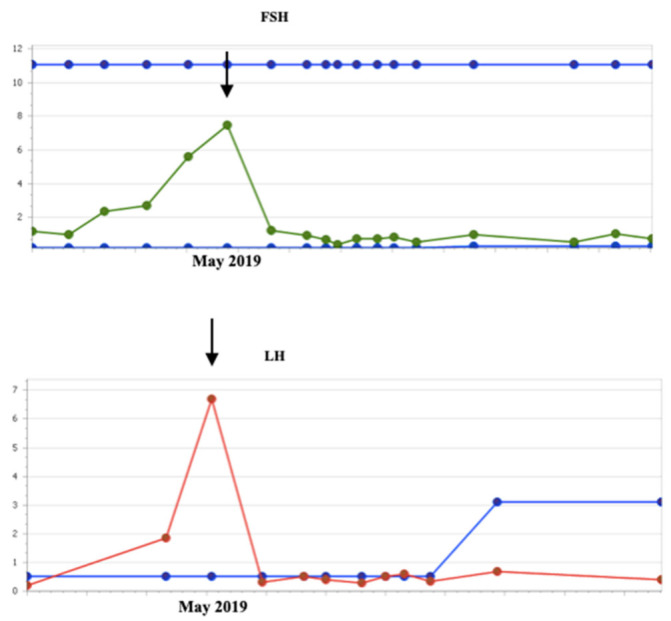
Evolution of FSH and LH values before and after starting hormone treatment (July 2019).

**Figure 3 children-10-00115-f003:**
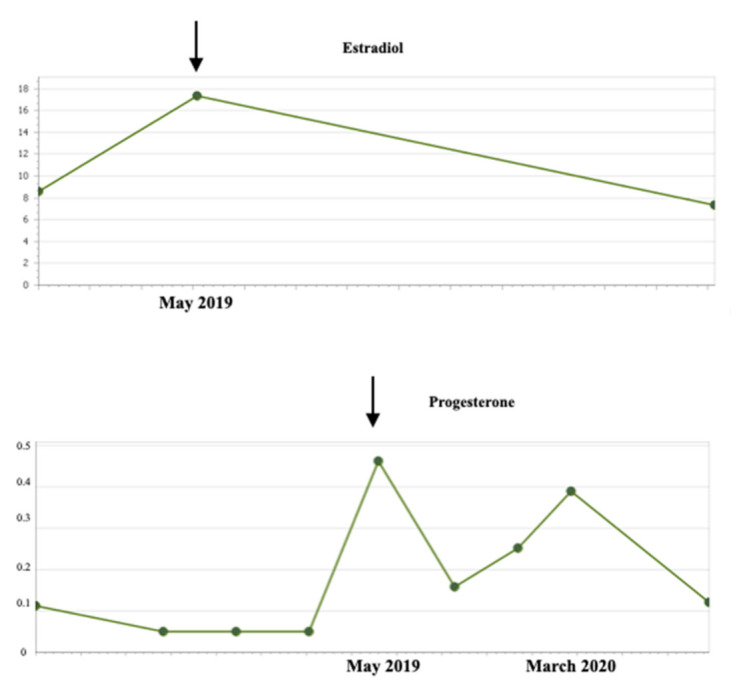
Evolution of estradiol and progesterone values before and after starting hormone treatment (July 2019).

**Figure 4 children-10-00115-f004:**
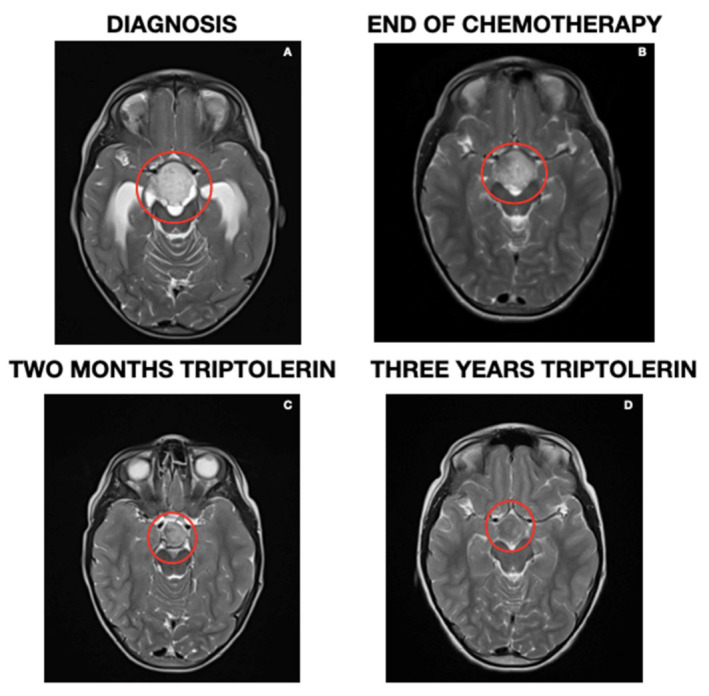
Magnetic resonance image at four different moments: at debut (**A**), end of treatment (**B**), after two months of hormonal treatment (**C**), and current image (**D**).

## Data Availability

The datasets generated during/or analyzed during the current study are available from the corresponding author on reasonable request.

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
