# Peer review of "Effect of the Use of Gnrh Analogs in Low-Grade Cerebral Glioma"

_children, 2023, doi:10.3390/children10010115_

Round 1
Reviewer 1 Report
Overall a very interesting and thought-provoking case report.
Significant English language editing required particularly in abstract, introduction and case report. Discussion reads very well.
Consider expanding line 46-56 regarding previous publications regarding steroid hormones and effect on CNS tumors - agree with authors that detailed discussion is best placed at beginning of discussion section in manuscript but could optimally set the stage at the end of the introduction.
Line 158-162 recommend including references.
Also recommend including clinical changes associated with tumor growth - were there any symptoms that improved following the treatment?
Again, a very interesting and relevant report, very thought provoking.
Author Response
Dear reviewer,
First of all, thanks for all your suggestions. My answers:
- English: I have reviewed again all the article in order to improve english redaction.
- Expanding line 46-56: I have done it.
- Line 158-162: I have included references.
- Including clinical changes associated with tumor growth: I have included a sentence that explain it.
You can see the changes in orange. Moreover, you can see there are some sentences in blue. The journal asked me to expand the article, so those parts are news.
Reviewer 2 Report
The authors presents a case of a female child who presented with a suprasellar pilomyxoid astrocytoma (grade 1) at age 2 years and 1 month. The tumor lacked estrogen and progesterone receptors by immunohistochemistry on pathology exam. She was treated with vinblastine stable disease. Her follicular stimulating hormone and lutenizing hormone increased in at some point in her treatment that was unclear in the history, with clinical evidence of precocious puberty. This was treated with triptorelin, a gonadotropin releasing hormone with shrinkage of the tumor on this medication that is not seen within vinblastine.
The discussion describes the authors’ hypotheses with regard to the etiology of the tumor shrinkage.
On line 117, the wording “despite the practical normalization of the FSH and LH levels, it continues with striking growth rate, requiring dose adjusted to 128 mcg/kg/dose every 21 days.” The authors need to clarify regarding what had the striking growth rate.
In figure 2, the X-axes of the 2 graphs have different starting dates, one being in November of 2017 and the other in March of 2018. This gives the first impression to the reader that the FSH and LH peaked at the same time. In the FSH graph, what is the top line that is straight for? Why was it not in the LH graph? In both graphs, the relatively straight line at the bottom of the graphs was not identified. The authors should identify what each line on each graph is for. Likewise, in figure 3, the dates on the X axes of both graphs should be normalized between the estradiol and progesterone graphs.
A timeline of the patient's presentation and subsequent treatment should be better detailed. The interval between the date of diagnosis and the timeframe in which the hormonal abnormalities were detected is unknown. The time interval in which the triptorelin was started was unclear. Failure for was well done. However, there is no particular need for the office to use the brand name (Decapeptyl) for triptorelin rather than the generic, as this is distributed under different brand names.
The authors needs to identify potential causes of the tumor response other than the triptorelin in the discussion. The discussion and conclusions to also identify how patients may be identified with for this form of treatment for future clinical trials.
Author Response
Dear reviewer,
First of all, thanks for all your suggestions. My answers:
- English: I have reviewed again all the article in order to improve english redaction.
- Line 117: I have included a sentence later in order to explain it.
- Figure 2: I have changed it.
- Timeline: I have included dates.
- Decapeptyl: I have removed this word and I replaced it with triptorelin.
- Potential causes: I have included other possibilities explaining our results.
You can see the changes in green. Moreover, you can see there are some sentences in blue. The journal asked me to expand the article, so those parts are news.
Round 2
Reviewer 1 Report
Overall much improved and a very interesting case which adds insight into the literature.
Reviewer 2 Report
All concerns were adequately addressed by the authors.